# Biological Control Strategies and Integrated Arthropod Pest Management for *Camellia oleifera*

**DOI:** 10.3390/insects16121244

**Published:** 2025-12-09

**Authors:** Yifei Xie, Jinxiu Yu, Wan Deng, Shaofeng Peng, Chi Li, Xuanye Wen, Wuhong Zhong, Mi Li

**Affiliations:** 1Institute of Forestry and Grassland Protection, Hunan Academy of Forestry, Changsha 410018, China; xyf8779@163.com (Y.X.);; 2Yuelushan Laboratory, Changsha 410018, China; 3State Key Laboratory of Utilization of Woody Oil Resource, Hunan Academy of Forestry, Changsha 410004, China; 4Hunan Shennongguo Oil Eco-Agriculture Development Co., Ltd., Changsha 410000, China; 5Center for Biological Disaster Prevention and Control, National Forestry and Grassland Administration, Shenyang 110034, China

**Keywords:** natural enemies, microbial pesticides, ecological regulation, Theaceae, oil-tea

## Abstract

This review synthesizes biological control and Integrated Pest Management (IPM) strategies for *Camellia oleifera* arthropod pests. We comprehensively document the natural enemy complex, key ecological interactions, and practical management methods. The paper provides a robust framework for sustainable, ecologically balanced pest control, highlighting the potential of emerging biorational technologies.

## 1. Introduction

*Camellia oleifera* (oil-tea) is a woody oil-bearing species native to East Asia, particularly China, where it plays a vital role in both agricultural production and ecosystem services. Cultivated mainly between 18° N and 34° N across more than 20 Chinese provinces. Since the 1990s, the global cultivation area has expanded from 2.4 to 5.2 million hectares (Mha), and annual tea oil production has tripled, reaching approximately 1.2 million tons [1]. Historically recorded over 2000 years ago, *C. oleifera* remains an economically significant species in China. Its oil, rich in unsaturated fatty acids, is widely used in culinary, cosmetic, and medicinal industries, thereby earning the accolade “Olive Oil of the East” [2].

The *C. oleifera* industry in China has expanded since the mid-20th century, with emphasis by the government on its strategic importance for providing edible oil security while advancing rural revitalization. Since the 1950s, national efforts to reclaim and restore plantations have steadily expanded the industry. By 2020, 4.451 Mha were under cultivation, with an annual production level of 720,000 tons valued at over 150 billion RMB. The Forestry and Grassland Industry Development Plan (2021–2025) aims to increase output to over 2 million tons on more than 6 Mha by 2025 [1,3,4].

Ecologically, *C. oleifera* is adapted to subtropical monsoon climates with red soil hills between 98° E and 121° E longitude. Its deep-rooted and evergreen canopy supports soil and water conservation, highlighting its environmental importance [5,6]. However, its widespread monoculture cultivation increases vulnerability to pests. A total of 392 damage-causing arthropod species have been recorded, including 386 insects and six mites across nine orders and 67 families, with Hemiptera and Lepidoptera as dominant orders (Table 1) [7,8,9,10,11,12,13,14,15,16,17]. Leaf- and stem-feeding pests constitute 72.8% and 19.2% of this fauna, while root, fruit, and flower pests comprise only 8% [18]. Pests include defoliators such as *Biston marginata*, borers like *Bacchisa atritarsis*, the seed weevil *Curculio chinensis*, and subterranean termites (*Odontotermes formosanus*) [10,19]. It is imperative to direct specific attention towards those pests characterized by exhibiting regional specificity or endemicity (Table 2) [14,18,20].

From the 1970s to the 1990s, pest control relied heavily on chemical pesticides, particularly organophosphates, resulting in resistance development, ecological disruption, and contamination issues. Since the 21st century, Integrated Pest Management (IPM) has gained prominence, emphasizing biological control, habitat management, and reduced-toxicity pesticides [21,22]. Biological methods, such as the use of native parasitoids, microbial agents, and predators, are now widely implemented, contributing to increased production of organic tea oil.

The widespread adoption of biological control strategies for this species is predicated upon extensive research conducted over the last three decades. This research has included comprehensive surveys and the identification of indigenous natural enemies such as microbial agents, parasitoids, and predators within *C. oleifera* agroecosystems. The biological characteristics of those enemies have been used in the formulation of effective application strategies for field deployment. Because both the tea plant (*Camellia sinensis*) and oil-tea (*C. oleifera*) belong to the genus *Camellia*, insect pests typically associated with tea are also likely to infest oil-tea. Ongoing investigations suggest a considerable overlap in the natural enemy groups associated with both host plants, indicating potential synergies in biocontrol research [10]. Furthermore, nucleic acid-based pesticides that leverage RNA interference (RNAi) technology represent an emerging frontier. These agents enable target-specific suppression of essential gene functions in oil-tea pests. Environmental benignity and minimal residual persistence contribute to their potential. This technology can mitigate the contamination of oil-tea by conventional chemical residues. As such, RNAi-based approaches are a potentially transformative, innovative research direction central to the development of green and sustainable pest management systems for oil-tea. This review comprehensively analyzes the current research advancements related to both biological control methods and the development of nucleic acid-based pesticides for managing oil-tea pests. Additionally, prospective avenues for future research within this critical domain are delineated.

## 2. Surveys and Taxonomy of Natural Enemies

The natural biodiversity within oil-tea agroecosystems is a foundational element critical to the effectiveness of biological control programs targeting pests of *C. oleifera*. Extensive research into the use of natural enemies has been undertaken by numerous Chinese investigators since the 1980s. These efforts have included systematic surveys and the taxonomic identification of natural enemies of oil-tea in the main oil-tea production regions. To date, these inventories have documented a rich natural enemy fauna associated with oil-tea pests, comprising over 600 species which include vertebrate predators such as various avian species [14,18].

### 2.1. Entomopathogens

Entomopathogens used in biological control encompass a range of microorganisms, including viruses, bacteria, fungi, and nematodes. These organisms typically exhibit potent pathogenicity towards insects and are capable of inducing a wide variety of diseases that frequently culminate in host mortality. To date, these entomopathogenic microorganisms comprise over 700 recorded species. However, despite this considerable taxonomic richness, only a limited subset of approximately 20 species has been successfully developed and used in pest management programs [23,24].

#### 2.1.1. Viruses

The investigation into viruses that infect insects started in the early 19th century, while research on this subject within China was initiated in the 1970s [25,26]. The identification of viruses specifically infecting insect hosts and so possessing the capacity to effectively regulate pest populations has led to the development and widespread adoption of several viral biopesticide formulations to date. Thus far, a total of 38 viral species associated with pests of oil-tea have been documented; the categories of their respective types are listed in Appendix A.

#### 2.1.2. Bacteria

*Bacillus thuringiensis* (Bt) represents the predominant species identified among the entomopathogenic bacteria isolated from pests afflicting *C. oleifera*. This bacterium has an abundance of varieties and strains that exhibit potent insecticidal activity particularly against larvae of various Lepidopterans. Consequently, numerous Bt varieties or strains have been successfully cultivated in vitro and subsequently have been used in agricultural production, including the strains designated as HD-1, 7216, and 140, which are used in China [27]. Specifically, within the context of oil-tea IPM research in China, in 1973, forestry scientists isolated B. thuringiensis subsp. galleriae from B. marginata, subsequently designated as strain “735” [28]. Furthermore, additional isolation and screening efforts have led to the identification of strains 111 and 119 from the tea leafroller (*Protapanteles theivorae*) and the oil-tea tussock moth, *E. pseudoconspersa*, respectively [20].

#### 2.1.3. Fungi

*Among microbial pathogens*, *entomopathogenic* fungi (EPF) *in general* represent the most speciose group. *They* are a crucial biological control resource for IPM of insect pest populations *and play* a significant role in the management of pests affecting oil-tea [29]. The isolation and identification of EPF species directly from oil-tea plantations, complemented by surveys of pathogenic fungi associated with shared pests on the tea plant, have led to the identification of 41 species of EPF associated with pests [18,27,30,31] (Appendix A). These fungal species are classified within four primary taxa (reflecting historical and current classifications): Ascomycotina, Basidiomycotina, Deuteromycotina, and Zygomycotina. Among the recorded EPF species, *Beauveria bassiana* and species within the genus *Metarhizium* are the most extensively used in practical applications to date. Researchers are actively engaged in the isolation, screening, and cultivation of these fungi, resulting in the development and characterization of numerous effective strains. Notable examples include *B. bassiana* strains designated as BbIII22, Bb01, and BbHA-11, as well as *Metarhizium* spp. strains Ma1775, Ma09, and FJMa201101 [32,33,34,35,36].

#### 2.1.4. Nematodes

Entomopathogenic nematodes (EPNs) are a specialized group of parasitic natural enemies targeting insects. With their active host-seeking behavior and parasitic mode of action, EPNs are particularly effective against pests in concealed environments, such as soil-dwelling and boring insects. Consequently, EPNs are recognized internationally as a unique class of environmentally compatible and highly effective biological insecticides [37]. Studies have primarily concentrated on evaluating the potential of using nematodes in the genus *Heterorhabditis* and the species *Steinernema carpocapsae* for the control of soil-inhabiting insect pests. Examples of target species in these studies include the tea seed weevil *C. chinensis* and the scarab beetle *Anomala corpulenta* [38,39].

### 2.2. Parasitoids

Investigations into the parasitic Hymenoptera associated with pests of *C. oleifera* began in the 1980s. Initial studies, primarily conducted in Hunan and Zhejiang provinces, documented six parasitoid wasp species in two important host insects: the lappet moth, *Lebeda nobilis*, and the Japanese felt scale, *Metaceronema japonica* [40,41,42]. Concurrently, an egg parasitoid in the family Scelionidae was identified attacking eggs of the shield bug, *Poecilocoris latus* [43]. Subsequent research focusing on parasitic natural enemies within oil-tea agroecosystems has largely adopted an ecological-level perspective. The principal goal of these more recent surveys has been to characterize the dominant pest species present and identify their associated natural enemy complexes [15,16,44,45,46,47,48,49,50]. These investigations have resulted in the documentation of 148 species of parasitoid wasps in 21 families of the Hymenoptera (Table 1). Among the parasitoid fauna taxa associated with oil-tea pests, the families Ichneumonidae, Aphelinidae, Braconidae, Encyrtidae, and Aphidiidae exhibited the highest species richness. These five families collectively account for 22.9%, 16.3%, 12.7%, 7.8%, and 4.2% of the total identified parasitoid wasp species, respectively (Figure 1A). Parasitic wasps associated with several prevalent and economically significant insect pests of oil tea are detailed in Appendix A.

Regarding parasitic Diptera, Wang and Wan [51] identified 12 species of Tachinid flies (Diptera: Tachinidae) within tea plantations whose known hosts include pest species affecting oil-tea. Integrating these findings with research by Chen, Zhang, and potentially others, a total of 18 tachinid species associated with oil-tea pests have been documented. The majority of these tachinids function as parasitoids primarily targeting Lepidopteran pests of *C. oleifera*. Notably, seven of these tachinid species have been confirmed to parasitize the oil-tea tussock moth, *E. pseudoconspersa*, during its larval or pupal stages [27,52].

### 2.3. Predators

The predatory natural enemies targeting insects and mites comprise both arthropod and vertebrate taxa. Comprehensive reviews have synthesized data from various studies documenting a diverse predator fauna [18,27,46,53,54,55]. This includes 336 species of predatory arthropods along with three amphibian, 12 avian, and one rodent species. The primary prey spectrum for this predator complex consists mostly of pests in the orders Lepidoptera and Hemiptera. Other frequently-targeted prey species include smaller insect taxa such as thrips and weevil larvae. The details regarding the identified vertebrate predator species are shown in Appendix A.

#### 2.3.1. Predatory Insects

Investigations into predatory natural enemies within oil-tea agroecosystems began in the 1960s. Shen and Shen [56] documented that the lady beetle *Chilocorus rubidus* effectively controlled populations of the Japanese felt scale, a significant pest of *C. oleifera*. Subsequent studies identified another lady beetle, *Hyperaspis sinensis*, as a principal predator of Japanese felt scale [57,58]. More recently, surveys focusing specifically on predatory insects were carried out in 2015 within oil-tea plantations located in the Yunnan and Hunan provinces, documenting a diverse predatory insect fauna, totaling 51 species belonging to 15 families across eight insect orders: Coleoptera, Dermaptera, Diptera, Hemiptera, Hymenoptera, Mantodea, Neuroptera, and Odonata. The observed prey spectrum for these predators primarily encompassed aphids, Lepidopteran larvae, scale insects, and various other small insects [46,54].

The present study accounts for 240 predatory insect species (Table 1), distributed across 10 orders and 33 families. The five families with the highest representation in terms of species number are Coccinellidae, Carabidae, Syrphidae, Reduviidae, and Chrysopidae, contributing 31.7%, 17.5%, 11.3%, 5%, and 4.6% to the total, respectively (Figure 1B).

#### 2.3.2. Spiders and Predatory Mites

Spiders are the numerically dominant predatory arthropods in many agricultural settings, including *C. oleifera* agroecosystems. Surveys in Hunan oil-tea plantations (1997–1998) documented 89 spider species across 20 families [53]. Comparative analysis with data from tea garden spider assemblages indicated that about 25% of the spider species identified in oil-tea plantations have not been reported from tea gardens. This ecological difference might be linked to the structure of oil-tea plantations, particularly the dominance of canopy-dwelling spider sub-communities. The overall community structure and stability of spiders may be influenced by the denser understory vegetation often present in these plantations.

While no specific surveys of predatory mite populations in oil-tea plantations exist, Zhang and Tan [27] identified seven predatory mite species from four families that prey on *C. oleifera* pests. These included *Amblyseius nicholsi*, which has been observed to feed on *C. oleifera* pollen following overwintering [59]. Future research should document the occurrence, population dynamics, and survival of these beneficial mites within oil-tea agroecosystems. Despite limited current data, predatory mites, with their predatory habits and associations with pests or resources like pollen, represent a promising, largely unexplored resource for biological control strategies against oil-tea pests. Figure 1C shows the percentage distribution of species numbers. Appendix A provides a comprehensive catalogue of all identified spider and predatory mite species relevant to this context.

## 3. Implementation of Biological Control and Development of Integrated Management Strategies

Biological control of *C. oleifera* pests began in China in the 1960s, focusing on native natural enemies and their ecological roles. Recent advances have integrated biological control into comprehensive IPM systems, combining attractant-based methods, resistant cultivars, and pest forecasting. These strategies enhance natural enemy conservation, provide effective pest suppression, and improve yield stability in oil-tea cultivation

### 3.1. Application of Natural Enemies

#### 3.1.1. Viruses

Among the viruses used for controlling pests relevant to *Camellia* species, two have achieved large-scale application, particularly within tea plantations, and have been officially registered as biological control agents in China. These are the NPVs of the *Euproctis pseudoconspersa* (EpNPV) and the NPV of *Ectropis obliqua* (EoNPV) [60,61]. Beyond these established viral agents, extensive research, notably by Tang, et al. [62], has focused on the formulation and application strategies for the NPV isolated from the nettle caterpillar, *Iragoides fasciata* (IfNPV). Laboratory bioassays allowed the determination of the optimal concentration for IfNPV application to be 1 × 10^7^ PIBs/mL. Furthermore, studies investigated synergistic effects when combined with Bt, identifying an optimal Bt concentration of 2000 International Units (IU)/µL for the mixture. Subsequent field trials at three distinct geographical locations involved applying this IfNPV-Bt admixture against successive generations (1st, 2nd, and 3rd) of *I. fasciata*. A mortality rate of 100% was achieved against the first-generation larvae. However, mortality rates observed for the second and third generations were somewhat lower at about 88% [62,63].

Several other entomopathogenic viruses targeting pests also relevant to oil-tea have been deployed within tea plantation management programs. These include the NPV of *Buzura suppressaria* (BsNPV), the Granulosis Virus of *Andraca bipunctata* (AbGV), and the GV of *Adoxophyes orana* (AoGV). Ye, et al. [64] provided detailed information regarding the application and characteristics of these viruses, and therefore, will not be reiterated in the present review.

#### 3.1.2. Bacteria

While widely used elsewhere, *Bacillus thuringiensis* (Bt) application for *C. oleifera* pest control remains largely limited to small-scale trials. Early investigations using Bt strain “735” against the oil-tea geometrid (*B. marginata*) showed promising results, achieving over 95% mortality by day 7 [28]. However, standalone Bt applications against other significant *C. oleifera* pests have been less effective, yielding only 33.83% population reduction for *C. patrona* [65] and approximately 50% efficacy for *Dasychira baibarana* [66].

These findings suggest limitations for Bt as a sole control agent, highlighting the need for integrated approaches. Synergistic effects have been explored, such as co-application with viral agents like IfNPV (as previously mentioned). Further supporting this, He, et al. [67] isolated two synergistic bacterial strains, *Paenibacillus azoreducens* strain X-11 and *Staphylococcus* sp. strain X-2, from the nettle caterpillar *Thosea sinensis*. A mixture of these isolates with Bt achieved a maximum control efficacy of 67.20% against *T. sinensis* in tea plantations, an 8.8% improvement over Bt alone [68].

#### 3.1.3. Fungi

*Beauveria bassiana* and *Metarhizium anisopliae* are the primary fungal biocontrol agents used against oil-tea pests. Studies show *B. bassiana* effectively controls Lepidopteran pests *Porthesia atereta* and *Andraca bipunctata*, achieving 70–80% mortality within nine days [35,69]. For Coleopteran pests, foliar *B. bassiana* sprays achieved 72% control of *Basilepta melanopus* adults [70], while soil incorporation yielded 59–71% control against *Myllocerinus aurolineatus* larvae and pupae [71].

*Metarhizium anisopliae* demonstrates efficacy against the leafhopper *Empoasca vitis*, achieving 74% control after 14 days with a single application [72]. Soil incorporation against tea seed weevil larvae resulted in 51% mortality after 30 days [36].

Integrating entomopathogenic fungi with other biocontrol agents or compatible pesticides significantly enhances control. Co-applying *B. bassiana* with the entomopathogenic nematode (EPN) *Heterorhabditis beicherriana* resulted in 100% mortality of *Anomala corpulenta* larvae [39]. Similarly, *B. bassiana* combined with *Steinernema carpocapsae* achieved 74.5% control of tea seed weevil larvae [38]. Joint application of *B. bassiana* and 1.5% pyrethrin also provided superior control against *E. vitis* [73].

To facilitate broader adoption, optimizing fungal production is crucial. Qiu, et al. [74] successfully increased *Aschersonia placenta* mycelial biomass by 3.6-fold and conidial yield by 10-fold through culture media optimization, significantly enhancing production feasibility for commercial application.

#### 3.1.4. Parasitoids and Predators

Compared to entomopathogenic microorganisms, research and field applications of macroscopic parasitoids and predators for oil-tea pest management are less extensive, often confined to small-scale trials.

Lei, et al. [75] assessed the parasitoid wasp *Sclerodermus guani* against the longhorn beetle *Chreonoma atritarsis*. Parasitoid-to-host release ratios of 1:1, 1:2, and 1:3 resulted in 50.9%, 67.3%, and 72.7% *C. atritarsis* mortality, respectively. However, *Dastarcus helophoroides* could not complete its life cycle on this host, precluding its use.

Huang [76] investigated *Aphidoletes abietis* (aphid midge) and *Harmonia axyridis* (lady beetle) against the tea aphid. After 14 days, their control efficacies were 57.5% and 66.9%, respectively. A combined release significantly increased efficacy to 75.4%, suggesting synergistic effects. Sun [77] observed high predation rates by *Chilocorus rubidus* and *Hyperaspis sinensis* lady beetles, with adults consuming over 85% of Japanese felt scale ovisacs, highlighting their potential for scale insect management in oil-tea.

### 3.2. Development of IPM Strategies for C. oleifera

#### 3.2.1. Selection of Pest-Resistant Camellia Cultivars

Host plant resistance, a core component of Integrated Pest Management (IPM), leverages genetic traits within cultivars to confer pest resistance or tolerance. These mechanisms include constitutive or induced production of deterrent secondary metabolites and morphological features that impede pest establishment or feeding. *C. oleifera* exhibits significant genetic diversity, leading to varied pest responses across genotypes. However, research explicitly investigating host plant resistance mechanisms against *C. oleifera* pests remains limited.

Notably, Zhao, et al. [78] found “Guangning red flower oil-tea” significantly less susceptible to tea seed weevil due to its thick pericarp, physically preventing oviposition and larval infestation. Complementing this, Li [79] observed “Changlin 2” displayed remarkably low fruit drop (14.96%) and weevil infestation (12.4%) rates induced by weevil activity compared to other cultivars. These findings suggest “Changlin 2” possesses inherent resistance to the tea seed weevil, though the precise underlying mechanisms require further investigation.

#### 3.2.2. Research and Application of Semiochemicals

Semiochemicals, especially attractants, offer valuable tools for IPM in *C. oleifera*. Research primarily focuses on the tea seed weevil: palmitic acid attracts males, while phytol attracts females [80]. Conversely, n-octadecane repels the weevil, and oleamide strongly attracts it [81].

Studies also show that semiochemicals influence beneficial insects. Zhong found nonanal, n-tridecane, and (E)-2-decenal elicit EAG responses in the egg parasitoid *Trissolcus halyomorphae*, with a preference for n-tridecane [82]. Intriguingly, pest sex pheromones can act as kairomones, attracting natural enemies. For example, a 1:9 ratio of nepetalactone and nepetalactol from the tea aphid’s sex pheromone strongly attracts lacewings (*C. sinica* and *C. septempunctata*) [83]. This highlights the potential for exploiting semiochemicals to enhance biological control.

#### 3.2.3. Physical Control Methods

Physical methods, like colored sticky traps and light traps, are crucial for *C. oleifera* pest management. Yellow sticky traps effectively capture pests like thrips, leafhoppers, and aphids [84]. However, their use requires careful consideration due to potential non-target captures, including beneficial natural enemies. For instance, yellow and green traps attract lacewings (*C. sinica* and *C. septempunctata*) [83]. Minimizing attractive colors to beneficials or designing non-sticky traps that attract natural enemies to trapped pests could mitigate this.

Blacklight traps (320–400 nm UV) are effective against nocturnal pests. Strategic placement at 1.5 m in open *C. oleifera* areas can significantly reduce fruit infestation. Wang, et al. [85] reported blacklight trapping reduced peach fruit borer (*Conogethes punctiferalis*) infestation from 28.78% to 10.72% over two years. Specific light wavelengths also show direct inhibitory effects. Qiao, et al. [86,87] found 520–525 nm and 590 nm LED light reduced survival, shortened development, and decreased fecundity in *Ectropis grisescens* larvae. Biochemical analysis revealed increased protective enzyme activity in adults exposed to these wavelengths [88]. However, blacklight traps also incur non-target effects on beneficial insects like fireflies, nocturnal pollinators and natural enemies, necessitating strategic deployment to minimize ecological impact [89].

Combining physical methods enhances efficacy. Hong, et al. [90] demonstrated that integrating sex pheromone lures for *Ectropis obliqua* with blacklight trapping achieved a 71% pest reduction, significantly outperforming either method alone.

#### 3.2.4. Forecasting Indicators and Systems

Effective pest management in *C. oleifera* relies on robust early warning systems. Lin, et al. [91] developed a predictive model for the tea seed weevil, integrating 36 indicators (weevil activity, environmental conditions, host phenology) into a composite risk index. Chen, et al. [92] identified six key ecological factors (canopy closure, aspect, planting density, and three others) influencing *Casmara patrona* outbreaks. They then constructed a highly accurate regression model (97% success rate) to forecast larval infestation probability and severity. These models exemplify progress in proactive pest management for *C. oleifera*.

### 3.3. Development and Prospects of Nucleic Acid-Based Pesticides (RNAi Pesticides)

Exogenous RNA interference (RNAi) represents a transformative frontier for enhancing *C. oleifera* pest management. Unlike broad-spectrum chemical pesticides, nucleic acid-based pesticides utilize double-stranded RNA (dsRNA) to silence essential genes in target pests with high specificity, thereby offering a “biorational” alternative that minimizes harm to non-target organisms, including natural enemies [93]. While research on *Camellia* pests is still emerging, studies on the polyphagous pest *Spodoptera litura* (common cutworm), which also infests Theaceae, have demonstrated the efficacy of targeting genes related to pheromone perception (Orco, OBPs) [94,95], detoxification (P450s, GSTs) [96,97,98,99], and reproduction [100,101]. Crucially, recent advancements have begun translating these findings to specific oil-tea pests. For instance, silencing the Notch gene in the tea leaf beetle, *Basilepta melanopus*, significantly impaired ovarian development and reduced fecundity [102]. Although challenges regarding formulation stability and production costs remain, RNAi technology holds significant promise for integration into IPM programs. By reducing reliance on chemical insecticides, RNAi strategies can potentially preserve the populations of the extensive natural enemy complexes documented in *C. oleifera* agroecosystems, promoting a more sustainable ecological balance.

## 4. Conclusions

Effective *C. oleifera* cultivation demands sophisticated pest management. China has documented 392 pest species and over 600 natural enemies, including pathogens and nematodes, within its agroecosystems. Biological control, via natural enemy releases and microbial biopesticides like *B. bassiana*, is increasingly integrated into IPM frameworks alongside host resistance, forecasting, and physical methods.

Concurrent advances in RNAi pesticide technology, driven by new target gene discoveries and cost-effective dsRNA production, offer significant IPM promise. However, expanding *C. oleifera* cultivation, with diverse cultivars and environments, highlights regional IPM variations and the need for diverse natural enemy complexes. Continuous knowledge updates and sustained innovation in biological control are crucial for organic certification and sustainable oil-tea industry growth.

## 5. Policy Implementation Recommendations

China contributes over 90% of the global *C. oleifera* production, with an annual seed yield exceeding 2 million tons and a comprehensive industry chain valuation surpassing 160 billion CNY (22 billion US dollars). As a unique strategic woody oil-producing resource endemic to China that possesses both significant ecological value and economic potential, strategic investment in the scientifically sound management of *C. oleifera*, particularly integrated pest control, promises substantial dividends. These include bolstering food and edible oil security in China, fostering economic transformation in mountainous regions, and enhancing China’s international standing in the woody oil-producing crop sector. Accordingly, this section outlines important recommendations with the goal of advancing *C. oleifera* pest management strategies in China towards alignment with international best practices.

First, research related to macroscopic natural enemies and the application of control measures should be prioritized. Because predatory and parasitic arthropods represent the most diverse guilds of natural enemies within *C. oleifera* agroecosystems, investment in this field of study is crucial for identifying novel beneficial species and optimizing their practical application through augmentative biological control. Addressing documented instances of suboptimal field efficacy (e.g., ~20% control of Japanese felt scale by *C. rubidus* [56]) requires advanced mass-rearing technology to enable sufficiently high initial release densities. Furthermore, research into kairomonal responses (e.g., to pest pheromones) and visual attractants (specific colors attractive to beneficial species [83] offers a promising avenue for developing “attract-and-retain” strategies. Such approaches could involve the targeted deployment of specific attractants or colors post-release to enhance the establishment, persistence, and overall control efficacy of introduced natural enemies.

Second, advanced microbial control and RNAi technologies should be considered. Because microbial control is currently the most widely applied biological control method, continued basic research on entomopathogenic viruses, bacteria, and fungi is essential to improve formulation stability, environmental resilience, persistence, and virulence, potentially using enhanced techniques via genetic engineering [103]. Concurrently, RNA interference (RNAi) technology presents significant opportunities for biological control of pests. Exploring synergistic strategies, such as RNAi targeting of host immunocompetence genes designed to enhance microbial biopesticide efficacy [104,105], holds promise. However, overcoming challenges related to cost-effective, large-scale dsRNA production remains important for the commercial viability of nucleic acid-based pesticides.

Third, judicious pesticide application strategies that are compatible with natural enemy conservation need to be developed and implemented. Addressing the current knowledge gaps will require research into synergistic frameworks that minimize effects on non-target species and preserve ecosystem self-regulation. This involves advancing precision application technologies, such as UAVs equipped with variable-rate application systems and sensors for targeted treatment; this should be coupled with optimizing application parameters such as droplet control and timing to avoid peak natural enemy activity, potentially guided by micro-environmental modeling. Crucially, this requires researchers to prioritize the selection of environmentally benign, low-toxicity pesticides which should be combined with targeted delivery techniques (e.g., stratified canopy application) to reduce non-target exposure. Ultimately, establishing dynamic action thresholds that integrate both pest incidence and natural enemy population data is a major part of avoiding unnecessary interventions and maintaining the ecological balance in ecosystems, thus creating a “natural enemy-aware” chemical application system that ensures efficacy while preserving ecological functions.

While genetic engineering to enhance host plant resistance against key pests is technically conceivable, significant consumer concerns and market barriers associated with genetically modified food products present substantial commercialization risks for *C. oleifera* oil. Therefore, investment in genetically modified oil-tea development specifically for pest resistance is considered inadvisable under current conditions. Future re-evaluation may be warranted if public perception and regulatory policies evolve over time. As a vital woody oil-producing crop, the sustainable management of *C. oleifera* pests using environmentally sound strategies remains paramount. Despite challenges, cumulative advancements in IPM, particularly in natural enemy conservation and biopesticide development, engender optimism. The next fifty years are poised to witness further iterations of safer, more efficient, ecologically integrated control solutions, supporting the green transformation and sustainable intensification of the oil-tea industry through IPM.

## Figures and Tables

**Figure 1 insects-16-01244-f001:**
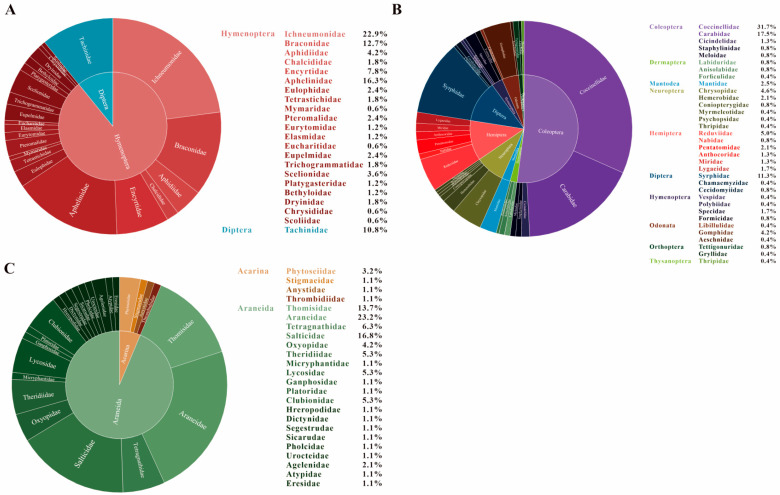
Family structures of parasitoids and predators associated with oil-tea pests in recorded China. (**A**): Parasitoids; (**B**): Predatory insects; (**C**): Spiders and predatory mites. The displayed data shows the proportion of species counts by family.

**Table 1 insects-16-01244-t001:** List of pest, parasitoid and predator orders and their family and species number in oil-tea gardens in China.

Class/Order	Family Number			Species Number			Species Proportion (%)		
	Pests	Parasitoids	Predators	Pests	Parasitoids	Predators	Pests	Parasitoids	Predators
**Insecta**									
Lepidoptera	28			155			39.55		
Coleoptera	9		5	49		125	12.50		37.20
Hemiptera	20		6	153		29	39.04		8.64
Orthoptera	5		2	14		3	3.58		0.89
Phasmida	1			1			0.27		
Isoptera	1			6			1.54		
Thysanoptera	1		1	4		1	1.03		0.32
Hymenoptera	1	21	4	2	148	8	0.52	89.16	2.39
Diptera	1	1	3	2	18	30	0.52	10.84	8.94
Odonata			3			13			3.89
Dermaptera			3			5			1.49
Mantodea			1			6			1.79
Neuroptera			5			20			5.97
**Arachnida**									
Acarina	3		4	6		7	1.54		2.09
Araneida			20			89			26.49
Total	70	22	57	392	166	336	100	100	100

**Table 2 insects-16-01244-t002:** Insect and mite pests of oil-tea in four growing regions of China.

Tea-Growing Region	Common Species	Damage Site	Main Region-Own Species (Damage Site)	Damage Site
The South China region	**Hemiptera:** ** Poecilocoris latus* **Lepidoptera:** *Casmara patrona* *Biston marginata* ** Euproctis pseudoconspersa* Strand *Parametriotes theae* **Coleoptera:** * *Curculio chinensis* * *Basilepta melanopus*	YL L L L L Fr YL	**Orthoptera:**	
			*Brachytrupes portentosus* Licht	SS
			**Lepidoptera:**	
			*Ectopis excellens* Butler	L
			**Hymenoptera:**	
			*Odontotermes formosanus*	R
			**Coleoptera:**	
			*Aeolesthes induta*	S
The Central and East China region			**Hemiptera:**	
			*Toxoptera aurantii*	F
			*Mctaccronemajaponic*	SS
			*Ricania speculum*	F
			*Eurostus validus*	L
			**Coleoptera:**	
			*Chreonoma atritarsi*	S
			**Lepidoptera:**	
			*Clania minuscula*	L
			*Phossa fasciata*	L
			*Latoia consocia*	L
			*Linocstis gonatias*	L
			**Hymenoptera:**	
			*Dasmithius camellia*	L
The Southwest region			**Hemiptera:**	
			*Pseudaulacaspis cockerelli*	YL
			*Chrysocoris grandis*	S, L
			**Coleoptera:**	
			*Anoplophora elegan*	S
			*Chreonoma atritarsi*	S
The Northwest region			**Hemiptera:**	
			*Dialeurodes citri*	F
			*Ceroplastes rubens*	F
			*Leucaspis japonic*	F
			**Lepidoptera:**	
			*Lebeda nobilis*	L
			**Coleoptera:**	
			*Anomala corpulenta*	S
			*Eurythrus blairi*	S
			*Chreonoma atritarsi*	S

* means these insect or mite species are the most serious pests in all of China or the whole special region. F: flushes (bud and young leaves); Fr: fruit; L: leaf; R: root; S: stem; SS: seedling stem; YL: young leaf.

## Data Availability

No new data were created or analyzed in this study. Data sharing is not applicable to this article.

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
