# Peer review of "Biological Control Strategies and Integrated Arthropod Pest Management for Camellia oleifera"

_insects, 2025, doi:10.3390/insects16121244_

Round 1
Reviewer 1 Report
Comments and Suggestions for Authors
Manuscript insects-4001230 by Xie et al. reviews options for biological control of insect pests of oil-tea and their integration into IPM programs. The article is written well and will be useful for many readers, especially those working on pest control in oil-tea and tea. It seems to have good coverage of available literature on the subject. However, I do not work in that system and cannot tell if any relevant references are missing.
One major problem that, in my opinion, needs to be addressed is the authors’ decision to include a section on RNAi while largely ignoring other pesticides. RNAi pesticides may be biorational, but they are not biological control agents. I suggest omitting this section and focusing on true biological control that uses living natural enemies. RNAi section could be developed into a separate publication.
If the authors insist on keeping the RNAi section in this article, it needs to be revised. Current heavy focus on cutworms does not make much sense. Interactions between RNAi pesticides and natural enemies need to be discussed in more detail. This discussion also needs to be expanded to include other pesticides and their place in IPM programs.
In addition, I have several more specific comments:
Lines 14-19. This Simple Summary does not describe the paper. It looks like it was mistakenly transferred from a different article during typesetting.
Line 55 and throughout the manuscript. C. oleifera and other Latin names need to be consistently italicized.
Lines 55-66 should be merged into a single paragraph.
Line 66. Need to quote publications that were used to compile Table 2 like it was done for Table 1.
Lines 114-115. Twenty species developed into biological control agents is actually very impressive.
Lines 137-139. Does this refer to insect pathogens in general, or to oil-tea pathogens?
Figure 1 is difficult to read because of very small size of the font. Perhaps pie charts can be replaced with bar graphs. That will allow stretching segments to larger sizes.
Lines 271-275. These two sentences are redundant.
Lines 358-365. Blacklight trapping may have serious non-target effects. This needs to be mentioned, like it is mentioned for yellow sticky traps above.
Lines 370-376. How is this related to biological control of tea-oil pests? I would imagine that weeds may serve as alternative hosts to some of them while providing alternative resources to some of the natural enemies. This needs to be discussed.
Lines 386-518. This section seems to be out of place in this manuscript. RNAi is not biological control. It could be a part of IPM, but so are other pesticides. Also, it is not clear why most examples are on cutworms. This technology is currently targeted against beetles. The two currently available products target corn rootworm and Colorado potato beetle.
Lines 567-581. Pesticide applications were not really discussed in the text. Therefore, I do not think it is appropriate to put them in Conclusions because these are supposed to follow from the information presented in the main body of an article.
Author Response
Thank you for the reviewer's valuable comments. Regarding your question/concerns, my responses are as follows:
Comments 1: Lines 14-19. This Simple Summary does not describe the paper. It looks like it was mistakenly transferred from a different article during typesetting.
Response: The simple summary has been rewritten.
Comments 2: Line 55 and throughout the manuscript. C. oleifera and other Latin names need to be consistently italicized.
Response: All the Latin scientific names has been verified and revised.
Comments 3: Lines 55-66 should be merged into a single paragraph.
Response: Paragraphs have been revised.
Comments 4: Line 66. Need to quote publications that were used to compile Table 2 like it was done for Table 1.
Response: The corresponding citations have been included.
Comments 5: Lines 114-115. Twenty species developed into biological control agents is actually very impressive.
Response: We thank the reviewer for their positive feedback on this point.
Comments 6: Lines 137-139. Does this refer to insect pathogens in general, or to oil-tea pathogens?
Response: The sentence was intended to convey that EPFs, which in general are the most speciose group of microbial pathogens, also represent a crucial biocontrol resource specifically within the context of C. oleifera IPM. To eliminate any ambiguity, we have revised the sentence in the manuscript to clearly separate the general statement from the specific application to oil-tea. Thanks.
Comments 7: Figure 1 is difficult to read because of very small size of the font. Perhaps pie charts can be replaced with bar graphs. That will allow stretching segments to larger sizes.
Response: We thank the reviewer for the valuable feedback on the readability of Figure 1. We agree that the font size was suboptimal. In response, we have significantly increased the font sizes both within the pie charts and in the figure legend to enhance clarity. While we appreciate the suggestion to use bar graphs, we have chosen to retain the pie charts as we believe they more intuitively and effectively convey the proportional composition of natural enemy families (i.e., the relative contribution of each family to the total), which is a key message of this figure. We are confident that the revised version, with improved typography, is now much easier to read while preserving the intended visual representation of the data.
Comments 8: Lines 271-275. These two sentences are redundant.
Response: The redundant sentence has been revised.
Comments 9: Lines 358-365. Blacklight trapping may have serious non-target effects. This needs to be mentioned, like it is mentioned for yellow sticky traps above.
Response: Following the reviewer's suggestion, we have added a new sentence/paragraph in the revised manuscript (Lines 368-370) to explicitly acknowledge the potential non-target impacts of blacklight traps and to recommend strategic measures for their mitigation. This change ensures a more balanced and ecologically responsible discussion of physical control methods.
Comments 10: Lines 370-376. How is this related to biological control of tea-oil pests? I would imagine that weeds may serve as alternative hosts to some of them while providing alternative resources to some of the natural enemies. This needs to be discussed.
Response: We fully agree with reviewer and have deleted this part.
Comments 11: Lines 386-518. This section seems to be out of place in this manuscript. RNAi is not biological control. It could be a part of IPM, but so are other pesticides. Also, it is not clear why most examples are on cutworms. This technology is currently targeted against beetles. The two currently available products target corn rootworm and Colorado potato beetle.
Response: We appreciate the reviewer’s insightful comment regarding the distinction between RNAi and classical biological control. While we agree the original section was overly detailed regarding Spodoptera litura, we believe outlining RNAi as an emerging "biorational" tool is vital for a comprehensive IPM outlook. Accordingly, we have significantly compressed this section into one concise paragraphs (approx. 170 words). Furthermore, the content of this section was moved from Section 4 to 3.3 and is no longer presented as an independent section. We believe this revision addresses the concerns about scope and relevance while retaining a forward-looking perspective.
Comments 12: Lines 567-581. Pesticide applications were not really discussed in the text. Therefore, I do not think it is appropriate to put them in Conclusions because these are supposed to follow from the information presented in the main body of an article.
Response: We thank the reviewer for raising the important point that discussions should be grounded in the main text. We respectfully wish to clarify that the section on pesticide application strategies is located in '5. Policy Implementation Recommendations' and not in the Conclusions section. The Policy Recommendations are intended precisely as a forward-looking platform to propose future research and implementation frameworks, which by their nature extend beyond the specific data reviewed in the main comments. Nevertheless, we fully agree with the reviewer's underlying principle.
Reviewer 2 Report
Comments and Suggestions for Authors
The manuscript “Biological Control Strategies and Integrated Pest Management for Camellia oleifera” (authors: Yifei Xie, Jinxiu Yu, Wan Deng, Shaofeng Peng, Chi Li, Xuanye Wen, Wuhong Zhong, Mi Li) is dedicated to the current status of research on pest management of C. oleifera as well as future research trajectories intended to provide for sustainable development of the C. oleifera industry. As presented, the MS induces a number of questions, namely:
- Lines 59-60: “A total of 392 arthropod pest species has been recorded, including 386 insects and six mites across nine orders and 67 families, with Hemiptera and Lepidoptera as dominant orders (Table 1) [7-17]". It is known that pests are those species that cause economically significant damage, i.e. not all those that do trauma (injury) deserve the status of a pest. Therefore, it makes sense to refer to insects that do not cause economically significant damage not as “pests”, but as species that cause damage.
- Lines 62-64: “Economically significant pests include defoliators such as Biston marginata, borers like Bacchisa atritarsis, the seed weevil Curculio chinensis, and subterranean termites (Odontotermes formosanus) [10, 19]". Here the authors have to introduce the phrase “economically significant pests” (see remark No. 1), which is incorrect, because pests are objects that cause economic damage (see remarks no. 1).
- Line 67: In Table 2, it is not good to indicate the damage site immediately after the name of the harmful species in parentheses, as it is common to indicate the name of the author of the original description of the species. If the species has been transferred to another genus, the author's name should also be indicated in parentheses. It is customary to use the letter L. to abbreviate the author of the original description, C. Linnaeus, and the letter F. to Johannes Christian Fabricius. In other words, in order not to mislead readers of the MS, parentheses should be replaced, for example, with slashes or square brackets. Additionally, the *icon (meaning these insects or mites are the most significant pests in China or a specific region) should be placed after the species name rather than before it.
- Line 193: “The predatory natural enemies targeting pests and mites comprises both arthropods”. It wasn't pests that should have been mentioned here, but insects.
- Line 369-376: Section 3.2.4 Weed Management 369 seems completely unnecessary in the review, as it is dedicated to weed control and has nothing to do with pests.
- Table S1 has a column called "Virus family". This column contains two names of virus genera (Cypovirus and Picornavirus), but not the names of the families ending in viridae. Therefore, it should be titled "Virus genus". And here is the abbreviation P. lepida (presumably Parasa lepida from the Eucleidae family, more commonly known as Limacodidae). It is necessary to provide the full generic name because the genus name is given in full at the first mention of the Latin name of an object.
- In Table S3, there are no headings on the right side of the table (presumably Host insects/development stages and Parasitic wasp species).
- There are no column headers in Table S5.
Author Response
Thank you for the reviewer's valuable comments. Regarding your question/concerns, my responses are as follows:
Comments 1: Lines 59-60: “A total of 392 arthropod pest species has been recorded, including 386 insects and six mites across nine orders and 67 families, with Hemiptera and Lepidoptera as dominant orders (Table 1) [7-17]". It is known that pests are those species that cause economically significant damage, i.e. not all those that do trauma (injury) deserve the status of a pest. Therefore, it makes sense to refer to insects that do not cause economically significant damage not as “pests”, but as species that cause damage.
Response: Thanks for your comments. We have already revised the inappropriate terms.
Comments 2: Lines 62-64: “Economically significant pests include defoliators such as Biston marginata, borers like Bacchisa atritarsis, the seed weevil Curculio chinensis, and subterranean termites (Odontotermes formosanus) [10, 19]". Here the authors have to introduce the phrase “economically significant pests” (see remark No. 1), which is incorrect, because pests are objects that cause economic damage (see remarks no. 1).
Response: We have revised the relevant inappropriate terminology.
Comments 3: Line 67: In Table 2, it is not good to indicate the damage site immediately after the name of the harmful species in parentheses, as it is common to indicate the name of the author of the original description of the species. If the species has been transferred to another genus, the author's name should also be indicated in parentheses. It is customary to use the letter L. to abbreviate the author of the original description, C. Linnaeus, and the letter F. to Johannes Christian Fabricius. In other words, in order not to mislead readers of the MS, parentheses should be replaced, for example, with slashes or square brackets. Additionally, the *icon (meaning these insects or mites are the most significant pests in China or a specific region) should be placed after the species name rather than before it.
Response: Thank you for your professional suggestions. We have redesigned Table 2 and added two new columns describing the damaged site, which should eliminate any ambiguity.
Comments 4: Line 193: “The predatory natural enemies targeting pests and mites comprises both arthropods”. It wasn't pests that should have been mentioned here, but insects.
Response: We have already revised the inappropriate terms.
Comments 5: Line 369-376: Section 3.2.4 Weed Management 369 seems completely unnecessary in the review, as it is dedicated to weed control and has nothing to do with pests.
Response: We have deleted this part of sections.
Comments 6: Table S1 has a column called "Virus family". This column contains two names of virus genera (Cypovirus and Picornavirus), but not the names of the families ending in viridae. Therefore, it should be titled "Virus genus". And here is the abbreviation P. lepida (presumably Parasa lepida from the Eucleidae family, more commonly known as Limacodidae). It is necessary to provide the full generic name because the genus name is given in full at the first mention of the Latin name of an object.
Response: Thank you for your professional suggestions. We have revised this part of tables.
Comments 7: In Table S3, there are no headings on the right side of the table (presumably Host insects/development stages and Parasitic wasp species).
Response: The headings have beed added.
Comments 8: There are no column headers in Table S5
Response: Column headers have been added in Table S5.
Reviewer 3 Report
Comments and Suggestions for Authors
In this paper, the authors conduct a review on biological control and the use of natural enemies for insect pest control in Camellia oleifera. The information presented is relevant and covers a wide variety of reports on the use of insect pest control organisms. It also identifies important
information gaps and future research directions. Despite the above, several changes are necessary to make the work publishable. The most important ones are summarized below:
-The title is too broad. It creates the expectation that weeds and plant pathogens will be covered in detail. It is recommended to limit it to insect pests and remove the information on weeds, which, in any case, is covered much less extensively than that related to insects or arthropods.
-The simple summary does not correspond to the content of the work.
-Line 30 (Abstract): This text does not meet the conditions to be considered a systematic review. For further clarity see:for further clarity see: https://www.prisma-statement.org/
-Almost all scientific names are not in italics. This needs to be corrected throughout the text and references.
-Figure 1: Graph A does not show parasitoids as indicated in the legend, but predators. Graph B does not correspond to predators, but to parasitoids.
-Section 3.2.4: This topic requires extensive discussion that is not recommended in this paper. It is suggested that this topic be removed.
-Section 4 (RNAi pesticides): This section is not related to pest management in C. oleifera. It appears as if two reviews on different topics have been combined, since this section deals with the development of RNAi pesticides, primarily using studies on Spodoptera litura, without any connection to major pests of C. oleifera. This section of the text needs to be integrated with the rest of the manuscript.
The attached PDF contains numerous comments on specific aspects.

Author Response
Thank you for the reviewer's valuable comments. Regarding your question/concerns, my responses are as follows:
Comments 1: The title is too broad. It creates the expectation that weeds and plant pathogens will be covered in detail. It is recommended to limit it to insect pests and remove the information on weeds, which, in any case, is covered much less extensively than that related to insects or arthropods.
Response: Thank you for the your professional and valuable comments. We fully agree with your observation that the original title was too broad and might lead readers to expect a detailed coverage of weeds and plant pathogens. To address this issue, we have made the following revisions: 1. We have revised the title, Simple Summary, and Abstract to focus specifically on the biological control and IPM strategies for arthropod pests, which accurately reflects the main content of our review; 2. We have completely deleted all sections and information related to weeds from the manuscript.
Comments 2: The simple summary does not correspond to the content of the work.
Response: We have revised the simple summary.
Comments 3: Line 30 (Abstract): This text does not meet the conditions to be considered a systematic review. For further clarity see:for further clarity see: https://www.prisma-statement.org/
Response: We thank the reviewer for this important and accurate observation. We fully agree that our review does not strictly follow the methodological criteria (e.g., predefined search strategy, screening, and quality assessment) required for a formal systematic review as outlined by the PRISMA statement. Therefore, we have accepted the reviewer’s suggestion and modified the terminology throughout the manuscript. We have replaced the phrase "systematically reviews" in the Abstract (Line 30) and the Introduction with "comprehensively reviews" to accurately reflect the scope and methodology of our work.
Comments 4: Almost all scientific names are not in italics. This needs to be corrected throughout the text and references.
Response: We have checked and revised all the Latin scientific names.
Comments 5: Figure 1: Graph A does not show parasitoids as indicated in the legend, but predators. Graph B does not correspond to predators, but to parasitoids.
Response: We have revised Figure 1.
Comments 6: Section 3.2.4: This topic requires extensive discussion that is not recommended in this paper. It is suggested that this topic be removed.
Response: Yes, we have deleted this comment.
Comments 7: Section 4 (RNAi pesticides): This section is not related to pest management in C. oleifera. It appears as if two reviews on different topics have been combined, since this section deals with the development of RNAi pesticides, primarily using studies on Spodoptera litura, without any connection to major pests of C. oleifera. This section of the text needs to be integrated with the rest of the manuscript.
Response: We appreciate the your insightful comment regarding the distinction between RNAi and classical biological control. While we agree the original section was overly detailed regarding Spodoptera litura, we believe outlining RNAi as an emerging "biorational" tool is vital for a comprehensive IPM outlook. Accordingly, we have significantly compressed this section into one concise paragraphs (approx. 170 words). Furthermore, the content of this section was moved from Section 4 to 3.3 and is no longer presented as an independent section. We believe this revision addresses the concerns about scope and relevance while retaining a forward-looking perspective.
Round 2
Reviewer 1 Report
Comments and Suggestions for Authors
I do not have any further commnets. In my opinion, all my concerns have been addressed.
Reviewer 2 Report
Comments and Suggestions for Authors
The manuscript “Biological Control Strategies and Integrated Arthropod Pest Management for Camellia oleifera” (authors: Yifei Xie, Jinxiu Yu, Wan Deng, Shaofeng Peng, Chi Li, Xuanye Wen, Wuhong Zhong, Mi Li) has been significantly improved in response to the comments of reviewers. At present, the MS is quite suitable for publication, with the exception of Figure 1, which needs a major revision. The labels on this Figure are very difficult to read, and those applied to the radii of the circles are nearly impossible to interpret.
Reviewer 3 Report
Comments and Suggestions for Authors
The authors made all the requested modifications and provided substantive responses to the issues that required them. The succinct section regarding RNAi pesticides is now much better integrated into the overall scope of the manuscript. Therefore, I consider this version meets the requirements for publication.